# Total column ozone measurements by the Dobson spectrophotometer at Belsk (Poland) for the period 1963-2019: homogenization and adjustment to the Brewer spectrophotometer

Janusz W. Krzyścin[1], Bonawentura Rajewska-Więch[1], Janusz Jarosławski[1]

[1]Institute of Geophysics, Polish Academy of Sciences, Warsaw, 01-452, Poland

*Correspondence to*: Janusz W. Krzyścin (jkrzys@igf.edu.pl)

**Abstract.** The total column ozone ($TCO_3$) measurements by the Dobson spectrophotometer (serial No. 84) have been carried out at Belsk (51°50', 20°47'), Poland, since March 23, 1963. In total, ~115,000 intraday manual observations were made by December 31, 2019. These observations were performed for different combinations of double wavelength pairs in UV range (AD, CD) and observation types, i.e., direct Sun, zenith blue, and zenith cloudy depending on weather conditions. The long-term stability of the instrument was supported by frequent (almost every 4 yr.) intercomparisons with the world standard spectrophotometer. Trend analyses, based on the monthly and yearly averaged $TCO_3$, can be carried out without any additional corrections to the intraday values. To adjust this data to the Brewer spectrophotometer observations, which were also performed at Belsk, a procedure is proposed to account for: less accurate Dobson observations under low solar elevation, presence of clouds, and the sensitivity of ozone absorption to temperature. The adjusted time series shows that the Brewer-Dobson monthly averaged differences are in the range of about ±0.5%. The intraday $TCO_3$ data base, divided into three periods (1963-1979,1980-1999, and 2000-2019), is freely available at https://doi.pangaea.de/10.1594/PANGAEA.919378 (Rajewska-Więch et al., 2020).

## 1 Introduction

The monitoring of total column ozone ($TCO_3$) started in 1924 in Oxford (the United Kingdom) with prototype of the Dobson instrument (DI). Before the Second World War, there existed data records from 2 stations Oxford (DI serial No. 1) and Arosa (DI serial No. 2) archived in the World Ozone and Ultraviolet Radiation Data Centre. After the international geophysical year in 1958, the total number of the stations with routine $TCO_3$ increased up to about 50. The number of ozone observations increased sharply in the early 1980s following the discovery of the ozone hole in the Antarctic spring by Chubachi (1984) and Farman et al. (1985). The variability of the ozone layer was of particular interests, as ozone is the only gaseous component of the atmosphere that absorbs shortwave solar radiation in the UVB range (280-315 nm), which is harmful to all living organisms on Earth. Ozone depletion became a widely debated environmental issue that led to the signing of an international agreement, the Montreal Protocol (MP) 1987, to protect the ozone layer by reducing the production and use of the ozone depleting substances (ODS).

Until, the late 1970 the Dobson spectrophotometer was the only optical instrument for $TCO_3$ ground-based measurements. Monitoring of $TCO_3$ required a lot of man-power because the Dobson observations involve many manual activities, and the obtained $TCO_3$ value depended to some extent on the ability of individual observers to manipulate the instrument. The need to automate $TCO_3$ measurements resulted in the development of an automated instrument – the Brewer spectrophotometer. It was designed in Canada and after improvements in the 1970s, it was available for ozone monitoring in the early 1980s. However, differences between the Dobson and the Brewer records data were found. Kerr et al. (1988) suggested that this difference was due to temperature dependence of the ozone absorption coefficients ($O_3AC$). The Bass & Paur $O_3AC$ (Bass and Paur, 1985), which replaced the previous Vigroux (1967) coefficients, have been used operationally in the $TCO_3$ ground-based observation network since the early 1990s (Komhyr et al, 1993). Vanicek (2006) found that the difference could reach ~4%

when comparing the Brewer and Dobson monthly mean data taken in Hradec Kralove, Czech Republic, due to use of the Bass & Paur $O_3AC$ at fixed temperature. There were several attempts to recalculate the $O_3AC$ (Redondas et.al., 2014) and finally the recalculation proposed by Serduchenko et al. (2014) was recommended for the ground-based $TCO_3$ network. The application of these temperature-dependent absorption coefficients significantly reduced the artificial seasonality in the Dobson-Brewer differences to less than 1% (Redondas et al., 2014; Fragkos et al., 2015). However, the data from the Dobson network has not yet been recalculated using the new $O_3AC$, as the vertical temperature and ozone profiles in the stratosphere are required for the observation site.

The ozone issue is still relevant as the rate of ozone recovery expected from the regulations of the MP 1987 and its subsequent amendments was also driven by recent climate changes (e.g., Steinbrecht et al., 2017; Ball et al., 2018). In addition, a surprising increase in the concentrations of anthropogenic chlorofluorocarbons in the troposphere has been found in some regions related to ODS leakage from its long-term storing reservoirs (Lickley et al., 2020) and a return to industrial use of ODS (Montzka et al., 2018; Dhomse et al., 2019). The Antarctic ozone hole surprises with its variability from an extreme small extent in 2019 (Krzyścin, 2020) to extreme large one in 2020 (https://public.wmo.int/en/media/news/2020-antarctic-ozone-hole-large-and-deep?). Moreover, severe chemical losses occurred in the Arctic stratosphere in spring 2020 (e.g., Manney et al., 2020; Wohltmann et al., 2020). Therefore, it is still worth monitoring ozone with the Dobson spectrophotometer, which was designed and put into operation almost 100 years ago.

The Central Geophysical Laboratory of the Institute of Geophysics, Polish Academy of Sciences, at Belsk (51.84°N, 20.78°E), Poland, started monitoring atmospheric ozone ($TCO_3$ and the ozone vertical profile by the Umkehr method) on March 23, 1963. There are only two stations in Europe with longer time series, Arosa (since 1926, Staehelin et al., 1998) and Hradec Kralove (since 1961, Vanicek et al., 2012). The importance of the Belsk's $TCO_3$ time series results from the monitoring carried out regardless of the weather conditions, excluding only days with continuous rain or snow fall. Various efforts have been made to support data quality (e.g., Dziewulska-Łosiowa et al., 1983; Degórska and Rajewska-Więch, 1991) because less accurate zenith sky $TCO_3$ observations during cloud cover were only available on certain days, especially in late autumn and winter. This paper presents further steps in the homogenization of the Belsk's $TCO_3$ time series for the period 1963-2019 in perspective of the long-term variability of the atmospheric ozone and possibility of replacing the Belsk's Dobson with the state-of-the art Brewer spectrophotometer.

## 2 Data and Methods

### 2.1 The Dobson Spectrophotometer

The Dobson spectrophotometer was designed to measure $TCO_3$ by the technique of the differential optical absorption spectroscopy (Dobson, 1957). The $TCO_3$ values from Direct Sun (DS) observations are calculated based on the Beer's Law applied to spectral ultraviolet (UV) solar irradiances for selected wavelengths with strong and weak absorption by the ozone layer. These wavelengths are denoted as A (305.5/325.0 nm), C (311.5 /332.4 nm), and D (317.5/339.9 nm). The $TCO_3$ value is determined from linear combination of the logarithm of the ratios between A pair and D pair irradiances, called the double AD wavelengths pair algorithm, to reduce the effects of the aerosols scattering on calculated ozone. It is also possible to use C pair irradiances instead of the A pair, i.e., the so-called double CD pair, to obtain $TCO_3$ if the solar irradiance at the shorter wavelength of A pair is too weak because of a large ozone absorption under high air masses ($\mu$ >3) and/or during cloudy conditions.

Another option to calculate $TCO_3$ by the Dobson spectrophotometer is to use scattered sunlight from the zenith sky (ZS) at the same wavelengths as for the DS observations. In this case, the same algorithm is implemented but the result needs to be reduced by empirical charts. The charts proposed by Rindert (1973) were used throughout the Belsk observations for AD and CD

double wavelength pairs. The operational data archived in the PANGAEA data base was obtained using the Bass & Paur $O_3AC$

at fixed temperature of -46.3°C according the recommendation of Komhyr et al. (1993).

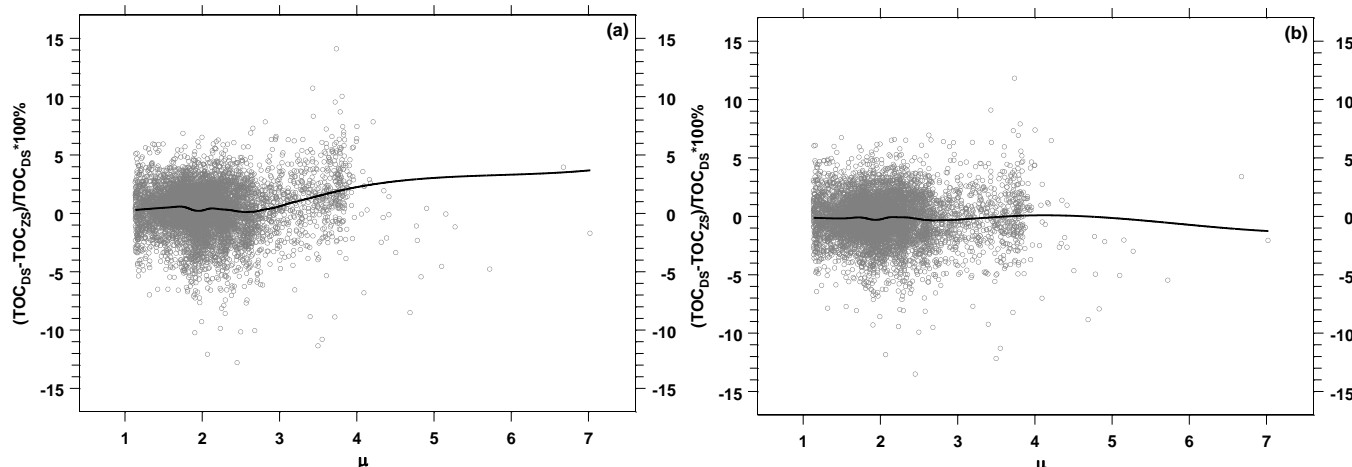

**Figure 1: Number of TCO₃ daily means based on the direct sun measurements for January and July in the period 1964-2019.**

Figure 1 shows time series of the number of daily TCO₃ means obtained from DS intraday observations with AD and/or CD wavelength pairs in January and July for the 1964-2019 period. The number of the DS daily means per month is larger in June because of usually less cloudy conditions this month compared to January and longer day length in summer (i.e., greater chance

for cloudless conditions). The monthly mean TCO₃ for January is mostly based on ZS observations. It is worth mentioning that reduced number of DS observations (less than 15 per month) was found in July for the period 1985-1995. During this period, ZS observations under a cloudless sky, i.e., observations of zenith blue (ZB) replaced DS observations because the solar disk was not clear due to heavy contamination of Belsk's atmosphere with industrial aerosols. The observers were then recommended to choose ZB measurement in the conditions of a cloudless sky.

**Figure 2: Air mass dependence of differences between daily mean TCO₃ values by DS and ZS measurements as a percentage of the DS TCO₃ values: original data (a), ZS TCO₃ after the correction using the transfer function (1) (b).**

The TCO₃ values by ZS observations could be calculated for almost all-weather conditions (excluding cases with heavy and variable cloudiness) but they are reliable, i.e., close to the DS values, for the air masses up to 2.8 (Fig.2a). The empirical reduction curves (Rindert, 1973), which were used to convert ZS values to equivalent DS values, did not reflect all possible

combinations of cloud types and their positions in the sky. Therefore, the DS-ZS differences can sometimes exceed 5%. To eliminate the drift of the DS-ZS differences, the following transfer function is used from the regression line fit to the relative differences between DS and ZS TCO₃ subsets for $\mu \subset [2.8, 4.0]$):

$$TCO_{3,ZS}* = (1 + 0.0125(\mu - 2.8)) \times TCO_{3,ZS} \quad \text{, for mi } \mu \subset [2.8, 4.0]$$

(1)

$$TCO_{3,ZS}* = 1.015 \times TCO_{3,ZS} \quad \text{, for } \mu > 4.0$$

where $TCO_{3,\,ZS}$ is the $TCO_3$ value derived from ZS observations, $\mu$ is air mass during the Dobson observation, and $TCO_{3,\,ZS}*$ is pertaining the corrected value for $\mu > 2.8$. However, the linear correction is not valid if $\mu > 4.0$ (Fig.2a). Only 0.7% of all $TCO_3$ observations were made at such high $\mu$ values. The fixed correction of 1.015 is applied if $\mu > 4.0$ but $TCO_{3,\,ZS}*$ values should be treated with caution. The smoothed pattern of the DS-ZS differences (Fig.2b) is close to zero after the application of the transfer function.

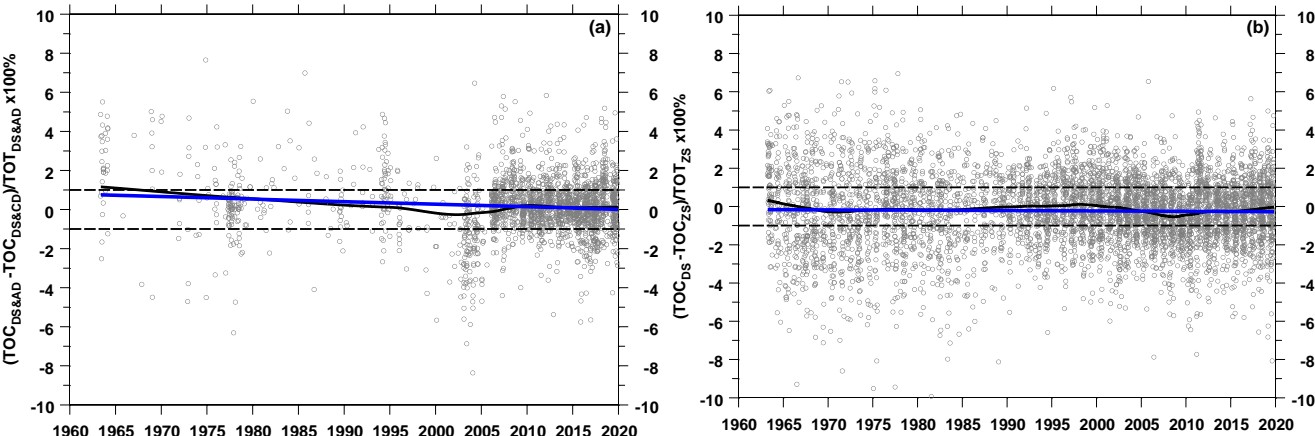

Figure 3: Time series of relative differences between daily means of TCO₃: DS&AD versus DS&CD measurements (a), DS versus ZS measurements (b). Black curve shows the smoothed pattern of the data. The blue line is the linear regression line.

DS&AD observations provide the most accurate $TCO_3$ values (Dobson, 1957) when the air mass is below 2.5. However, these observations are not possible for northern sites, such as Belsk, where the noon air mases above 2.5 appear in the period from 3rd November up to 9th February, and the maximum air mass at noon on 23th December is 3.75. Therefore, DS&CD observations are recommended for these periods (Dobson, 1957). Additional intraday $TCO_3$ values, especially after 2005, were calculated using the nearly simultaneous DS&AD and DS&CD observations to find a relationship between these $TCO_3$ values. Figure 3a shows that the relationship is quite stable in the 1963-2019 period, i.e., the smoothed differences (black curve) are within [0 %, 1 %] range and the linear regression line (blue line) is trendless. A similar comparison between DS and ZS observations, after application of the transfer function (1), provides that the smoothed differences are close to the zero line throughout the entire observation period (Fig. 3b).

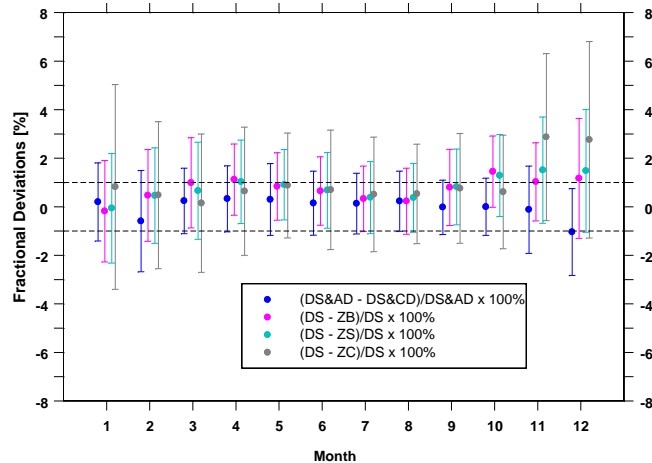

Figure 4: The monthly mean relative differences between daily TCO₃ means derived from different types of the Dobson measurements. Bar denotes the range: mean ± standard deviation.

The monthly mean $TCO_3$ differences between various subsets of the $TCO_3$ data over the period 1963-2019 are shown in Fig. 4. The differences are derived from the monthly averaging of the daily mean $TCO_3$ values for the following subsets: DS&AD, DS&CD, ZB, ZS, and for less reliable ZS observations, when the zenith was obscured by clouds i.e., the so-called zenith cloudy (ZC) observations. For all ZS intraday values if $\mu \geq 2.8$, the correction function (1) was applied. The $TCO_3$ differences

shown in Fig.4 are in the range of ± 1% almost all year round, excluding autumn/early winter months (October-November-December). Moreover, for each selected month, from January to September, the spread between the monthly mean differences is about ~1% for the all considered subsets of the Dobson observations. The maximum spread is 4% in December based on the difference of -1% for DS&AD vs DS&CD, and 3% DS vs ZC.

## 2.2 Calibrations of the Dobson spectrophotometer

The Dobson spectrophotometer serial No. 84 at Belsk has been operated continuously since March 23, 1963 in the same place with only a few breaks in observations (up to several weeks) due to the international calibration campaigns with other Dobson instruments. The frequent comparisons of the Belsk's Dobson with the world standard instrument in combination with the recalibration of the optical wedge of the instrument and the calculation of the resulting R/N tables were of key importance for maintaining the quality of the Belsk's Dobson. The R/N table is used to convert the dial reading (the so-called R value) obtained by the Dobson observer into the logarithm of the ratio between the light intensities in a pair of the UV wavelengths with weak and strong absorption by ozone (the so-called N value). N values are used in theoretical formulas to calculate $TCO_3$ (e.g., Dobson, 1957).

The first DI intercomparison was held at Belsk in June/July 1974. The participating Dobsons were compared against the World Primary Standard (WPS) Dobson Spectrophotometer No.83 and the optical wedge of the Belsk's Dobson was recalibrated (Dziewulska-Łosiowa and Walshaw, 1975). The next DI intercomparison was held in Potsdam (Germany) in June 1979 also with attendance of the same WPS. During the comparison the new R/N tables were obtained, which only slightly differed from those calculated earlier in the Belsk's intercomparison (Dziewulska-Łosiowa et al. 1983). The subsequent campaigns were in Arosa (1986, 1990, and 1995) with the WPS Dobson No. 83. Next intercalibrations (2001, 2005, 2009, 2014) were organized at the European regional Dobson calibration center, the Meteorological Observatory Hohenpeissenberg, Germany. The local European sub-standard, DI No. 64, was used as the reference instrument for comparisons in the center. After the intercomparisons, the new R/N tables were obtained and used immediately, i.e., after the instrument arrived at the observing site, as input to $TCO_3$ calculation algorithm. The retrieval software has not been changed since its first operational use in 1978 (Degórska et al., 1978). Table 1 summarizes the intercomparison campaigns with the Belsk's Dobson.

**Table 1. The intercomparison campaigns with the Dobson instrument from Belsk.**

| Site/Country | Year | Standard Instrument |
|---|---|---|
| Belsk/Poland | 1974 | World Standard. Dobson No.83 |
| Potsdam/Germany | 1979 | World Standard. Dobson No.83 |
| Arosa/Switzerland | 1986 | World Standard. Dobson No.83 |
| Arosa/Switzerland | 1990 | World Standard. Dobson No.83 |
| Arosa/Switzerland | 1995 | World Standard. Dobson No.83 |
| Hohenpeissenberg/Germany | 2001 | European Sub-Standard. Dobson No.64 |
| Hohenpeissenberg/Germany | 2005 | European Sub-Standard. Dobson No.64 |
| Hohenpeissenberg/Germany | 2009 | European Sub-Standard. Dobson No.64 |
| Hohenpeissenberg/Germany | 2014 | European Sub-Standard. Dobson No.64 |

The first homogenization of the Belsk's $TCO_3$ data for the period 1963-1981 by Dziewulska-Łosiowa et al., (1983) was based on the Potsdam's R/N tables, extraterrestrial constants determined at Belsk in 1974 during the first DI intercomparison, the $O_3AC$ by Vigroux (1967). The zenith sky irradiances (for AD and CD observations) were reduced to the DS equivalent values using the empirical charts according to Rindert (1973). The next homogenization was in 1991 by Degórska and Rajewska-

Więch (1991). All data collected before 1992 were re-evaluated on the reading-by-reading basis using the Bass&Paur $O_3AC$ (at fixed temperature at -46.3 °C) that were recommended for processing Dobson $TCO_3$ measurements since 1 January 1992.

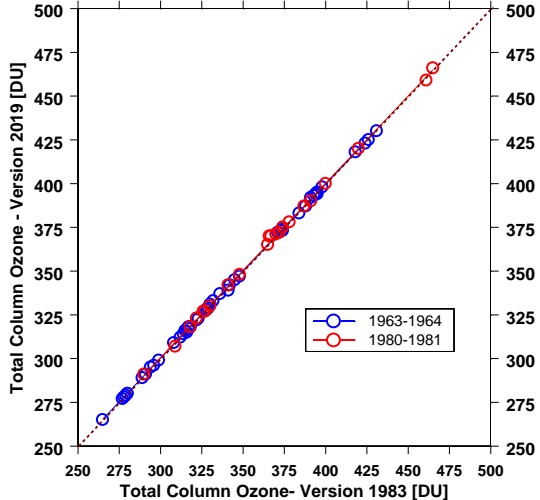

**Figure 5: $TCO_3$ daily means by AD&DS measurements in the period 1963-1964 and 1980-1981 calculated with present (2019) calculation algorithm (R/N tables based on the 2014 intercomparison) versus those by the same algorithm using old (1979) R/N tables.**

Optical wedge calibrations performed during the entire period of ozone observations at Belsk had only a slight effect on the $TCO_3$ values. Figure 5 shows the comparison of $TCO_3$ calculated for the first 2-year period (1963-1964) and last 2-year period (1980-1981) of the time series after the first homogenization in 1983. Original $TCO_3$ values from DS&AD observations presented by Dziewulska- Łosiowa et al. (1983), which were based on the Vigroux $O_3AC$, are converted to the values obtained using the Bass and Paur $O_3AC$ (i.e., multiplied by 0.9743, as recommended by the International Ozone Commission in 1992) and then compared with the corresponding values obtained from the current retrieval based on the Bass and Paur $O_3AC$ and R/N tables obtained in the last intercomparison in 2014. There is almost 1-1 correspondence between the two $TCO_3$ datasets, i.e., the slope of linear fit is 0.9999 and 0.9962 for the period 1963-1964 and 1980-1981, respectively. The maximum difference between $TCO_3$ values is less than 1 %. Small differences (<1%) are also found after comparison of DS&AD subset of $TCO_3$ values in August 2020 calculated using different R/N tables that were operationally used in the 1990s and 2020s (Fig.6).

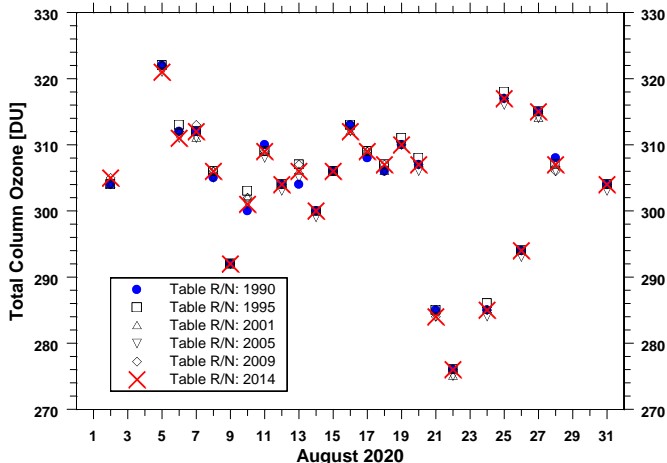

**Figure 6: Daily means of $TCO_3$ in August 2020 from DS&AD measurements as calculated with different R/N tables obtained during intercomparisons in the 1990s and 2000s (Table 1).**

From these comparisons of the DS&AD measurements, it can therefore be concluded that the Belsk's Dobson demonstrated stable performance throughout the entire measurement period. A linear approximation of $TCO_3$ values in periods between the calibration campaigns was not necessary to account for changes of R/N values and instrument aging.

**2.3 Adjustment to the Brewer spectrophotometer**

The ozone monitoring with the Brewer spectrophotometer serial No. 64 (BS64) Mark II (single monochromator) was launched at Belsk in 1991. Like other Brewers, the BS64 is a fully automated, self-testing, PC-controlled instrument intended for continuous, long-term observations in all weather conditions (e.g., Fioletov et al., 2005). The quality of BS64 measurements has been supported by regular (yearly or at least every two years) comparisons with the international travel reference instrument provided by the International Ozone Services Inc. (https://www.io3.ca/index.php). The BS64 instrument constants re-defined after each comparison were immediately incorporated into the operating retrieval.

The Brewer $TCO_3$ is derived from a weighted linear combination of solar irradiances at five wavelengths (306.3, 310.0, 313.5, 316.8, and 320.0 nm) to eliminate noise due to $SO_2$ absorption in the UV range (Kerr et al., 1988). The Brewer $TCO_3$ retrieval uses the Bass and Paur $O_3AC$ at -45 C° for two types (DS and ZS) of the observations. There are many daily BS observations regardless of weather conditions. Cycles of 5 observations, which are performed in ~3-4 minutes, are repeated every 5 to 20 minutes throughout the day depending on the instrument's schedule. The mean $TCO_3$ value, averaging 5 observations, is calculated if the scatter of these $TCO_3$ values is small. i.e., standard deviation is less than 2.5 Dobson unit (DU). The Dobson spectrophotometer provides instantaneous $TCO_3$ values, while the Brewer instrument gives the average of 5 observations, so this may be an additional source of differences between the spectrophotometers.

**2.3.1 Correction for the effective temperature**

Figure 7a shows the time series of the differences between Brewer and Dobson $TCO_3$ daily means (Brewer minus Dobson $TCO_3$ as % of the Dobson $TCO_3$ value) calculated for the period 2002-2019. This period was selected because more DS&AD and DS&CD observations were performed during the day (see Fig.3a) to construct a transfer function from the Dobson $TCO_3$ to the Brewer equivalent $TCO_3$ before the predicted change in monitoring policy to less frequent measurements during the day, i.e., only a few observations around noon in perfect weather conditions. The well-known seasonality of the Brewer-Dobson (B-D) differences, i.e., underestimation of the Dobson $TCO_3$ in the cold period of the year, could be observed after applying a low-pass filter, the locally weighted scatterplot smoothing (LOWESS, Cleveland, 1979), to the time series of B-D differences.

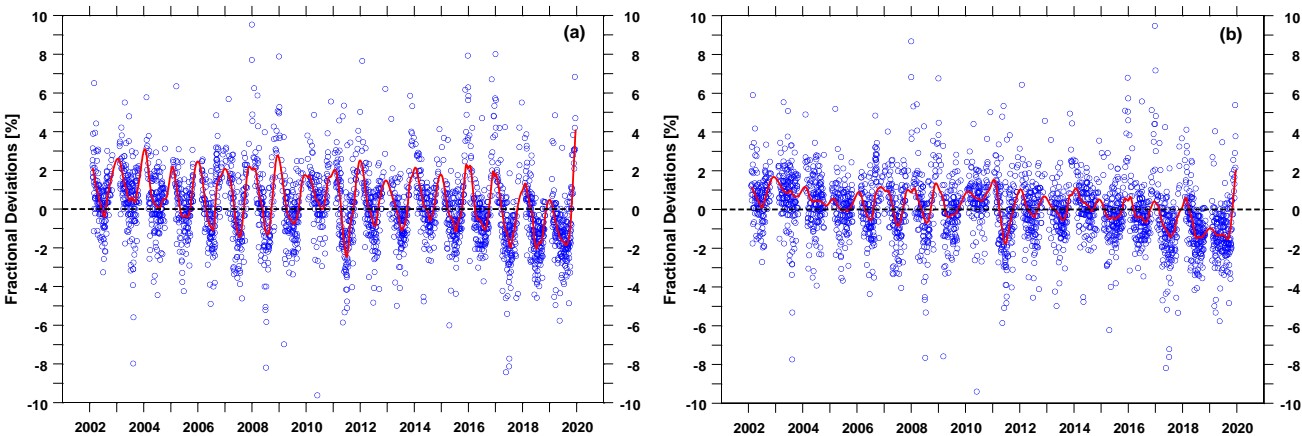

**Figure 7: The differences between the Brewer and Dobson TCO₃ daily means as a percentage of the Dobson TCO₃ value: original data after application of the correction function (1) to the Dobson ZS data (a), the Dobson and Brewer TCO₃ corrected for the effective temperature (b). Red curve represents the smoothed pattern by LOWESS filter.**

To reduce the B-D seasonal differences, the temperature dependent $O_3AC$ is applied instead of the operational $O_3AC$ assuming fixed temperature $T_0$, i.e., $T_0$= -46.3 °C (Dobson) and $T_0$= -45.0 °C (Brewer). Here we use the temperature correction factor, $\alpha$, calculated by Redondas et al. (2014), i.e., $\alpha$=0.104 % $K^{-1}$ (for Dobson) and $\alpha$=0.009 % $K^{-1}$ (for Brewer). These values are based on the $O_3AC$ dependence on temperature obtained by the Institute of Experimental Physics (IUP), University of Bremen (Serdyuchenko et al., 2014). The correction function to account for the actual effective temperature is as follows:

$$TCO_{3, NEW} = ( 1 + \alpha (T_0 - T_{eff}) ) \times TCO_{3, OLD} \tag{2}$$

where $T_{eff}$ is the effective (weighted by the ozone vertical profile) temperature, $TCO_{3,OLD}$ is the TCO$_3$ value at the fixed temperature, and $TCO_{3, NEW}$ is the temperature corrected value. Prior application formula (2), TCO$_3$ values by ZS Dobson observations need to be converted to the DS TCO$_3$ equivalent using transfer function (1). The effective temperature at noon is taken after the Tropospheric Emission Monitoring Internet Service (TEMIS) overpass data (http://www.temis.nl/climate/efftemp/overpass.html) based on the European Centre for Medium-Range Weather Forecasts (ECMWF) model estimates. Koukouli et al. (2016) found that the ECMWF effective temperature was in good agreement with that from the ozone sounding in northern hemisphere midlatitudes.

Figure 7b shows the time series of the B-D differences after applying the correction function (2). The visible seasonality in Fig.7a disappears, but there are still limited seasonal variations. To find out the range of these oscillations, the monthly mean B-D differences are calculated averaging the daily differences. The results are shown in Fig. 8a (DS subset) and Fig. 8b (ZS subset). A comparison of the original DS and ZS data (not accounting for T$_{eff}$ seasonal variability) provides that the seasonal B-D difference is of ~4.5% as calculated using the monthly mean difference values of ~3.5% and -1% for January and July (or August for ZS data), respectively. After application formula (2), the B-D difference is smaller, i.e., ~ 2.5% (DS) and 3% (ZS). The smoothed curved in Fig.7b provides that the Dobson TCO$_3$ values were ~1% lower (2002-2004) and ~1% higher (2018-2019) comparing with the Brewer values. Such discrepancies may be related to the Brewer ZS TCO$_3$ values, because they may be influenced by clouds (the Brewer ZS algorithm is based on a statistical relationship with parallel DS observations), which in some years result in overestimation (or underestimation) in relation to the Dobson TCO$_3$ values.

It is worth mentioning that the monthly B-D differences for the temperature corrected data are out of $\pm$ 1% range only in January and December (DS subset) and additionally in November for the ZS subset. To the authors' knowledge, the B-D differences for a less precise ZS subset have not yet been discussed by other authors. It seems that the stray light effect, which may be different in the Dobson and Brewer instruments (Belsk BS64 is a single monochromator spectrophotometer), is responsible for greater differences between these spectrophotometers during low solar elevation periods.

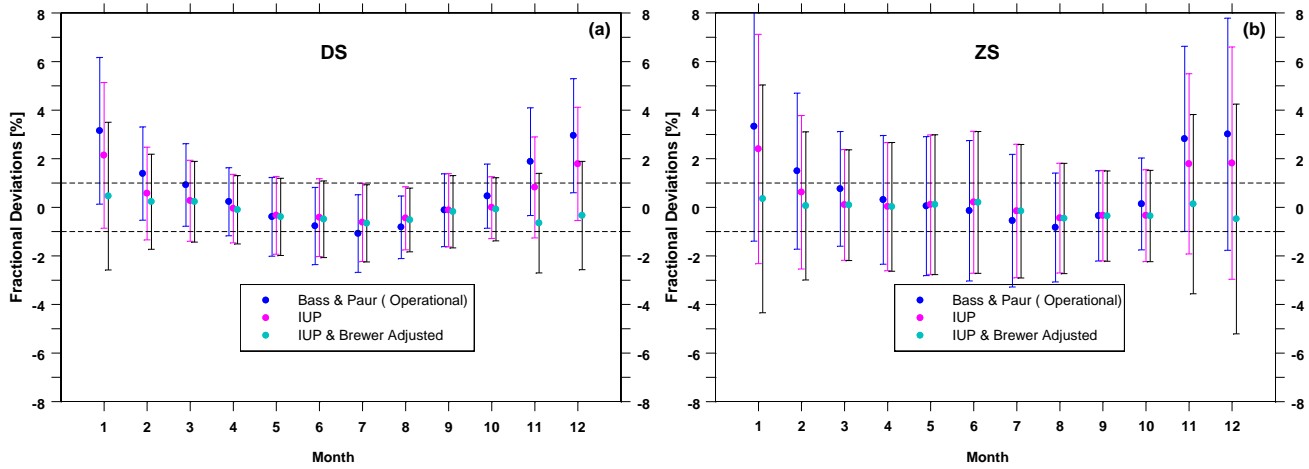

**Figure 8. The monthly means of the Brewer-Dobson differences for the operational data (Bass&Paur at fixed temperature), after application of the correction (2) for the dependence of ozone absorption coefficient on temperature based on the ozone absorption coefficients by IUP, and after adjustment of the Dobson TCO$_3$ to the Brewer equivalent by formula (3) accounting for the stray light effect differences between the spectrophotometers. The results are for: DS measurements (a), ZS measurements (b). Bars denote the range: mean ± standard deviation**.

### 2.3.2 Correction for the stray light effect

The stray light within the Dobson instrument causes an underestimation of the TCO$_3$ value by about a few percent, when slant TCO$_3$, $\mu \times TCO_3$, exceeds 800-900 DU (Evans et al., 2009). This underestimation also applies to the Brewer single monochromator spectrophotometers (e.g., Karppinen et al., 2015). Moeini et al. (2019) discussed the differences between TCO$_3$ values measured almost simultaneously by the Dobson and Brewer spectrophotometers due to the stray light effect. They found that the difference for low solar elevations (slant TCO$_3$ > 800 DU) was related to the instrument's individual

sensitivity to stray light, which may be particularly high for a single monochromator Brewer (Brewer Mark II), i.e., the same type operating at Belsk.

Here, it is decided to construct a simple correction function, which would shift all-monthly B-D differences for DS and ZS
subset (Figs. 8) to the smallest possible range. The following the Brewer-Dobson adjustment (BDA) function is proposed to transform $TCO_{3,\ NEW}$ (Dobson $TCO_3$ after application the transfer function (1) and the temperature correction by formula (2)) to its Brewer equivalent value denoted as $TCO_{3,\ DOB \rightarrow BRE}$ :

$$TCO_{3,\ DOB \rightarrow BRE} = (1 + \gamma \mu) \times TCO_{3,\ NEW}\ ,\ \mu > \mu_0$$

$$\hspace{9cm}(3)$$

$$TCO_{3,\ DOB \rightarrow BRE} = TCO_{3,\ NEW}\ ,\ \mu \le \mu_0$$

where constant values $\gamma$ and $\mu_0$ can be different for the ZS and DS subsets. The best pair of $\gamma$ and $\mu_0$ was experimentally derived examining many combinations of $\gamma$ and $\mu_0$, taking values from the range of [0, 0.02] and [2.5, 3.5], respectively. Finally, the following constants are derived: $\gamma_{DS} = 0.5625\ 10^{-2}$ and $\mu_{0,\ DS} = 2.95$ (for DS subset], and $\gamma_{ZS} = 0.6250\ 10^{-2}$ and $\mu_{0,\ ZS} = 2.8$ (for ZS
subset). The corresponding ranges of the monthly mean B-D differences are: [-0.66 % (July), 0.46 % (January)], and [-0.48 % (December), 0.35 % (January)], for the DS and ZS respectively. After application of the BDA function, the B-D monthly mean differences are almost in ± 0.5 % range (see Figs. 8.for the subset IUP & Brewer Adjusted)

## 2.4 Uncertainty of the Brewer adjusted Dobson $TCO_3$

Typically, daily $TCO_3$ averages were archived based on a few measurements around local with nominally best quality. AD&DS
observation shows the highest accuracy of all possible combinations of the double wavelength pairs (AD and CD) and observation type (DS, ZB, and ZS). This kind of the observation is not always possible because of weather conditions (clouds) and during low solar elevation. At the beginning of the $TCO_3$ observations at Belsk, a decision was made to increase the number of daily observations for selected days in each month in order to assess the uncertainty of $TCO_3$ observation. On such days, there were many, almost simultaneous observations with different instrument settings. For example, Figure 9 shows the
$TCO_3$ measurements at Belsk on August 8, 2017, planned to calculate differences between successive $TCO_3$ values. During the day, 46 observations were made for the following instrument settings: AD&DS, AD&ZS, and CD&DS.

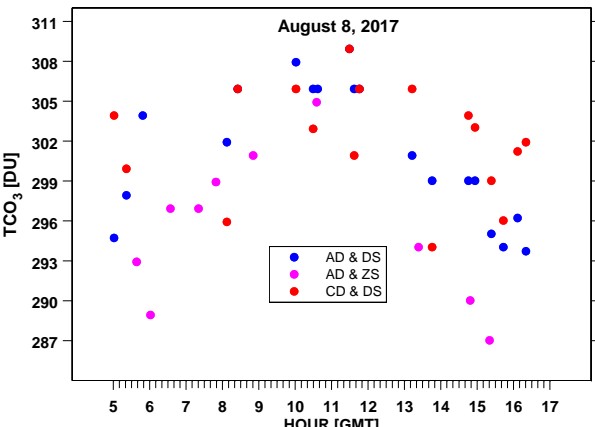

**Figure 9. The Brewer adjusted $TCO_3$ values measured at Belsk on August 8, 2017 for different settings of the Dobson spectrophotometer.**

The difference between two consecutive $TCO_3$ values sometimes exceeded 10 DU, even if the measurements were taken within
10 minutes. It is impossible for $TCO_3$ to change rapidly over such a short time scale taking into account natural variability of ozone in the stratosphere. The statistics of $TCO_3$ differences between the almost simultaneous Dobson measurement allows therefore to estimate the uncertainty of the Brewer adjusted Dobson $TCO_3$ values.

The statistics were obtained for various selected maximum ranges between the successive measurements, i.e. less than 1, 5, and 10 minutes. It was found that the statistical parameters did not depend on these ranges, which proved a reliable estimation

of the uncertainty. This uncertainty combines the instrumental uncertainty associated with the differences between various types of the Dobson measurements with the uncertainty resulting from the observer's skill to perform the measurements. Table 2 presents the statistics of the differences between $TCO_3$ values obtained almost simultaneously for different data subsets. The uncertainty of the Dobson observations for the subset with the nominal highest accuracy (DS&AD) is in the range of about [-1.15%, 1.09%] as derived from the [$5^{th}$-$95^{th}$] percentile range. The uncertainty increases with decreasing solar elevation and the largest value, [-4.58%, 4.04%], is for the ZS subset when air mass is between 3 and 4. The greatest uncertainty for this type of observations seems to be due to the cloud variability over the site as the ozone retrieval for non-DS observations is based on a statistical relationship with nearly parallel DS measurements, not taking into account the specific cloud configuration and optical properties.

**Table 2. The statistics of the differences between the Brewer adjusted Dobson $TCO_3$ values from two successive measurements taken at an interval of no longer than 10 minutes, for the period 1963-2019. The differences (in % of the mean of the two compared values) is shown for different sub-classes of the data and μ ranges. n denotes number of the measurement pairs used in the calculations**.

| Subset | Mean ± 1 SD | Median [$5^{th}$, $95^{th}$] | n |
|---|---|---|---|
| | $1 \leq \mu < 2$ | | |
| DS&AD | -0.00 ± 0.70 | 0.00 [-1.15, 1.09] | 399 |
| DS | -0.12 ± 1.25 | 0.00 [-2.15, 1.63] | 4070 |
| ZS | 0.01 ± 1.78 | 0.00 [-2.80, 2.98] | 5682 |
| ALL | -0.04 ± 1.58 | 0.00 [-2.62, 2.62] | 9752 |
| | $2 \leq \mu < 3$ | | |
| DS&AD | 0.02 ± 0.82 | 0.00 [ -1.10, 1.24] | 463 |
| DS | -0.18 ± 1.27 | -0.25 [-2.22, 1.80] | 3376 |
| ZS | -0.10 ± 2.11 | 0.00 [-3.40, 3.40] | 5102 |
| ALL | -0.13 ± 1.83 | -0.17 [-3.06, 2.93] | 8478 |
| | $3 \leq \mu < 4$ | | |
| DS&CD | 0.01 ± 1.39 | 0.00 [-2.26, 2.24] | 442 |
| DS | -0.32 ± 1.73 | -0.30 [-3.03, 2.27] | 826 |
| ZS | -0.11 ± 2.53 | -0.07 [-4.58, 4.04] | 1490 |
| ALL | -0.19 ± 2.28 | -0.12 [-3.90, 3.54] | 2316 |

**3 Results and Discussion**

The following subjects were considered to construct the Brewer adjusted record of the Belsk's Dobson measurements for the entire observation period (1963-2019): the focus was on DS&AD type of the Dobson observations during the intercalibration campaigns, DS&AD observations were well calibrated throughout the observation period (within ±1% range), the inter-comparisons were carried out in the warm period of the year (usually in summer), and there was a seasonal difference between the Dobson and Brewer spectrophotometers, depending to some extent on the effective temperature and the different response to the stray light in these instruments. It is worth mentioning that BS64 will ultimately be the primary ozone monitoring instrument at Belsk.

The first step in the data homogenization was the correction of Dobson's ZS observations, taking into account the drift in relation to DS&AD values (Figs.2). Application of the correction function (1) removed this shift (Fig.2b). Moreover, Fig.3a and Fig. 3b ensured that further correction to other types of the measurements was not necessary to eliminate long-term drift against the quality-controlled DS&AD subset. Thus, ZS-DS conversion charts proposed by Rindert (1973) were valid throughout the observation period, suggesting slight changes in the cloud characteristics at Belsk in the period 1963-2019.

Next step was to account for the temperature dependence of $O_3AC$ using the correction function (2) based on the very recent $O_3AC$ derived by Serdyuchenko et al. (2014). Application of this function removed part of the bias between the Brewer and Dobson spectrophotometers both for DS and ZS subset of the observations (Figs.7-8). The B-D monthly mean differences within ± 1% range occur in almost all months but not in months with low solar elevation at noon (November-December-January).

The last step of the homogenization was to eliminate the differences between the spectrophotometers found in periods of low solar elevation. These are probably related to the presence of stray light in spectrophotometers, causing underestimation of $TCO_3$ values at low solar elevations. The correction for the differences in the stray light effect in the spectrophotometers was proposed, see function (3), to reduce the B-D $TCO_3$ differences for low solar elevation. The stray light correction was not calculated separately for each instrument. However, Figure 10 shows, that the Brewer Adjusted $TCO_3$ values are only slightly sensitive to changes in slant $TCO_3$, i.e., within the max-min range between 0.99 and 1.01 derived from the smoothed profile of the ratio between $TCO_3$ values non affected (slant $TCO_3$ <800 DU) and affected by the stray light.

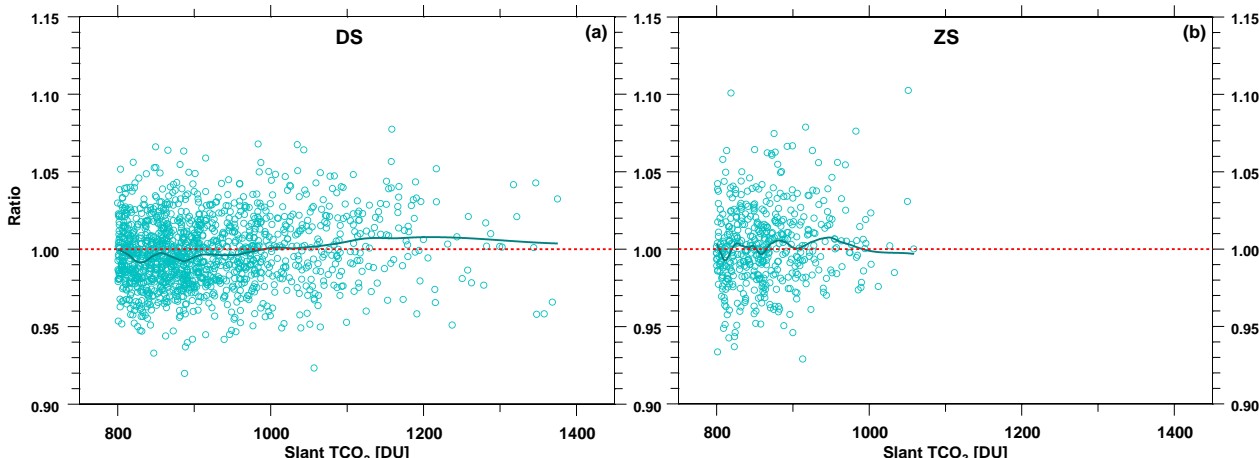

**Figure 10. The ratio between daily mean $TCO_3$ value based on the observations in the day with the slant $TCO_3$ smaller than 800 DU and daily mean $TCO_3$ comprising measurements with the slant ozone greater than 800 DU (points): The results are for: DS measurements (a), ZS measurements (b). The curve (in blue) represents the smoothed data.**

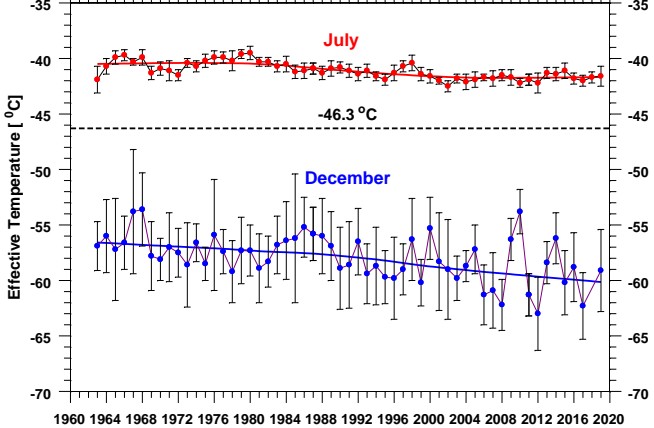

**Figure 11: Yearly means of the effective temperature at Belsk in December and July in the period 1963-2019.**

Table 3 provides the monthly mean values of the differences between the final and original $TCO_3$ values. The larger differences of about a few percentages are found in the late autumn and early winter months. For the warm season (April-September) and the corresponding higher solar elevation at noon, the differences are only in the range of ± 0.5 %.

Figure 11 shows that in December there was a significant decline in the effective temperature by ~ 4°C in the period 1963-2019 and a slight decline in July by ~ 1°C. For the original $TCO_3$ data (archived in the PANGAEA data base), such a decline caused the $TCO_3$ to be underestimated by ~0.4% and ~0.1% at the end of time series compared to the $TCO_3$ values at the start, in December and June, respectively.

Figure 12a and 12b show the long-term variability of the monthly mean $TCO_3$ for December and July, respectively, derived from the original data and the data after application of all correction functions (i.e., the Brewer adjusted Dobson $TCO_3$). The monthly means were calculated averaging the daily means from DS observations. If there were no such observations within the day, the daily means based only on ZS measurements were considered. The decline in the effective temperature (Fig.11) does not change the variability of the long-term $TCO_3$ pattern. A constant upward shift is found in the colder months (e.g.,

Fig.12a) and a slight downward shift in the warmer months (e.g., Fig.12b). A constant shift (upward) is also seen in the yearly mean values calculated as the average of all monthly means for each year (Fig.12c). Therefore, the trend estimates expressed in DU will be almost the same in original and corrected dataset if the trends are calculated as differences from the reference $TCO_3$ value (e.g., the $TCO_3$ monthly means before 1980).

**Table 3. The monthly and yearly statistics (mean, standard deviation, median, and the 5th-95th percentile range) of the differences between the Brewer adjusted Dobson $TCO_3$ and original ($O_3AC$ at fixed temperature) $TCO_3$ for the period 1963-2019.**

| Month/Year | n | Mean ± SD (%) | Median [ 5th, 95th] (%) |
|---|---|---|---|
| January | 1255 | 3.61 ± 0.81 | 3.68 [ 2.08,  4.78] |
| February | 1287 | 1.51 ± 0.82 | 1.39 [ 0.35,  3.15] |
| March | 1515 | 0.67 ± 0.34 | 0.63 [ 0.18,  1.26] |
| April | 1497 | 0.27 ± 0.23 | 0.25 [-0.07,  0.69] |
| May | 1577 | -0.09 ± 0.19 | -0.10 [-0.36,  0.21] |
| June | 1460 | -0.40 ± 0.16 | -0.42 [-0.61, -0.15] |
| July | 1594 | -0.53 ± 0.15 | -0.54 [-0.73, -0.33] |
| August | 1548 | -0.41 ± 0.18 | -0.44 [-0.67, -0.07] |
| September | 1520 | -0.03 ± 0.23 | -0.05 [-0.35,  0.35] |
| October | 1546 | 0.51 ± 0.31 | 0.47 [ 0.10,  1.02] |
| November | 1339 | 3.13 ± 1.16 | 3.40 [ 1.05,  4.71] |
| December | 1251 | 4.48 ± 0.53 | 4.54 [ 3.44,  5.24] |
| Year | 17389 | 0.93 ± 1.68 | 0.25 [-0.59,  4.55] |

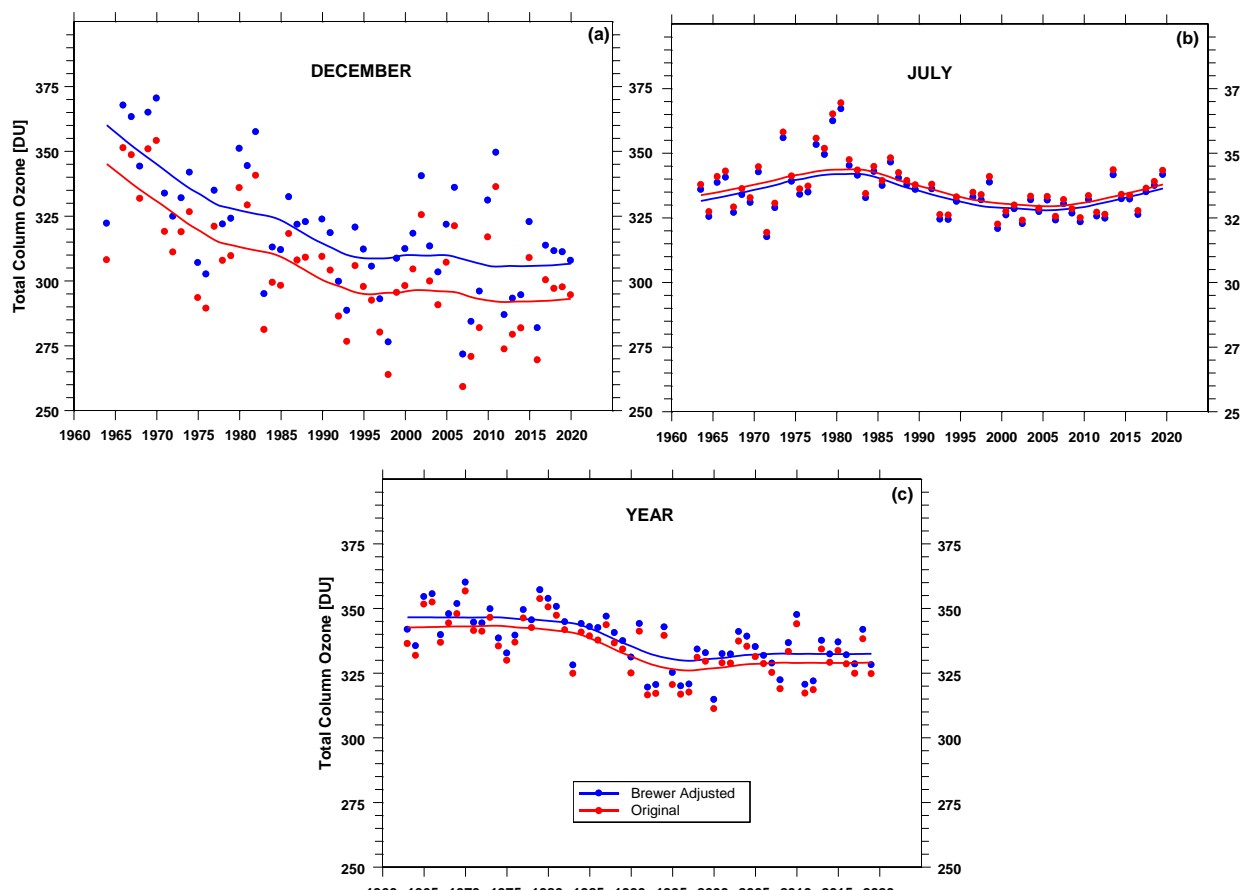

**Figure 12: Time series of the averaged daily $TCO_3$ means for Belsk in the period 1963-2019 using the original data (Bass&Paur at fixed temperature) and the Brewer adjusted Dobson data: monthly means in December (a), monthly means in July (b), yearly means (c). The red and blue curves show smoothed pattern for the original data and the Brewer adjusted Dobson data, respectively.**

### 4 Data availability

The dataset used in this article is available on the PANGAEA repository at https://doi.org/10.1594/PANGAEA.919378 (Rajewska-Więch et al., 2020).

## 5 Conclusions

The $TCO_3$ observations by the Dobson spectrophotometer at higher latitudes such as Belsk, with frequent non ideal conditions for the ozone observations, i.e., numerous cloudy days and low solar elevation at noon during the cold period of the year, should be subject of special quality checks prior to searching for the long-term trends and comparisons with satellite data. Frequent intercomparisons of the Belsk's Dobson spectrophotometer, which were carried out in perfect weather conditions, showed the stability of the instrument throughout the observation period for the most reliable DS & AD observations. However, such observations were not possible for many days, especially in the cold season of the year. The data correction procedure is proposed to account for: less accurate observations at low solar elevations, the presence of clouds, and the temperature sensitivity of ozone absorption.

The results of all intraday measurements for the period 1963-2019 were previously stored at the PANGAEA repository base with additional information including: time of observation (UTC), cloudiness type, air mass, and description of the wavelength pair and observation type selected for each individual measurement (Rajewska-Więch et al., 2020). The present analysis shows that the Brewer adjusted Dobson $TCO_3$ values are reliable for $\mu < 4$ or slant $TCO_3$ up to 1400 DU (Fig.10).

Corrections according to formula (1) and (3) are a linear function of air mass, so any user can easily calculate them based on archived air masses for all measurements. The effective temperature can be taken from the TEMIS data base and the correction for the ozone absorption dependence on temperature given by formula (2) also requires simple calculations. Thus, it will be possible to use the homogenized Dobson data for comparative studies with other $TCO_3$ data sources.

The present analysis shows that the original data (archived in PANGAEA data base) can be used in trend analyses based on the $TCO_3$ monthly and yearly means. Application of all proposed corrections to the original data provides that the resulting smoothed time series exhibits a constant shift (upward for the cold season and much less downward for the warm season) relative to the original long-term pattern. This finding supports the validity of previous trend analyses of the Belsk's Dobson data (e.g., Krzyścin and Rajewska, 2009; Krzyścin et al., 2013; Krzyścin et al., 2014). The resulting Brewer adjusted Dobson $TCO_3$ values are found in good agreement with the Brewer data. This will allow the construction of the merged Dobson-Brewer time series, as the Brewer observations will soon replace the Dobson measurements at Belsk.

**Author contributions.** JK wrote the paper and did statistical analysis, BR carried out the data processing, and JJ provided the Brewer data.

**Competing interests.** The authors declare that they have no conflict of interest.

**Acknowledgements.** The authors would like to acknowledge the support from the project Network of Geophysical Observatories of the Institute of Geophysics, Polish Academy of Sciences subsidized by the Polish Ministry of Education and Science (Decision No. 13/E-41/SPUB/SP/2020).

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
