# Peer review of "Total column ozone measurements by the Dobson spectrophotometer at Belsk (Poland) for the period 1963-2019: homogenization and adjustment to the Brewer spectrophotometer"

_Earth System Science Data, 2021_

## Author Comment (AC1)

**General comments:**

*The authors present one of the longest series of measurements of the total column of ozone globally. They also discuss the conditions and the procedures which ensure the high quality of the measurements. The scientific value of the presented dataset is high, and the manuscript is within the scope of the journal.*
*What I mainly miss, is a section wherein the authors would quantify the uncertainties of the final dataset. Uncertainty budget is of exceptional importance for anyone who would use the data. Thus, I strongly recommend that the authors should quantify the overall uncertainty and add the corresponding section.*
The new section is added " 2.3 Uncertainty of the Brewer adjusted Dobson TCO data" describing uncertainties in the data. (line. 260-285).

*A few more changes are also necessary prior to the publication of the manuscript. Specific comments are provided below.*
*Specific comments:*
**Below, in response to the reviewer's comments, we present the text with the suggested correction. The numbering of the lines relates to the corrected manuscript** .

*In the data files, or at least in an accompanying description file, please specify whether time is UTC or something else.*
" The results of all intraday measurements for the period 1963-2019 have been previously stored at the PANGAEA data base with additional information including: time of observation (UTC), cloudiness type, air mass, and description of the wavelength pair and observation type selected for each individual measurement (Rajewska-Więch et al., 2020)." (line 351-353)

*L7: please define that #84 is the serial number of the instrument.*
"The total column ozone ($TCO_3$) measurements by the Dobson spectrophotometer (serial No. 84) have been carried…. "(line 7)

*L13: please add "which were" before "also performed"*
"To adjust this data to the Brewer spectrophotometer observations, which were also performed at Belsk, a procedure is proposed to account for …." (line 13-14)

*L20: Please explain that #1 and #2 are the serial numbers of the instruments*
" there existed data records from 2 stations Oxford (DI serial No. 1) and Arosa (DI serial No. 2) archived in…"  (line 21-22)

*L22: "TCO" instead of "TOC" like in the rest of the document. At the same line, the authors probably mean that "the number of ozone observations increased sharply" instead of "The ozone observations were triggered".*
"$TCO_3$ increased up to about 50. The number of ozone observations increased sharply in the early 1980s …" (line 23-24)

*L33: Delete "the"*
"for ozone monitoring" (line 4)

*L36: which ground-based network?*
"have been used operationally in the $TCO_3$ ground-based observation network" (l.37-38)

*L42: Similar results to those reported by Redondas et al. (2014), have been also reported by Fragkos et al., (2015).*
"Redondas et al., 2014; Fragkos et al., 2015)"

*L47: In addition to Ball et al., the following study should be also cited: Steinbrecht, et al. (2017).*
", has also been driven by the recent climate changes (e.g., Steinbrecht et al., 2017; Ball et al., 2018)." (line 47)

*L50: Please add references for the Arctic ozone depletion in 2020. For example: Wohltmann et al. (2020); Manney et al. (2020); Inness et al. (2020).*
"Moreover, a severe chemical loss appeared in the Arctic stratosphere in spring 2020 (e.g., Manney et al., 2020; Wohltmann et al., 2020)." (line 52)

*L54: Delete "of the"*
"started monitoring atmospheric ozone" (line 56)

*L55: Delete "including"*
" there are only two stations with longer time series, Arosa (since 1926, Staehelin et al., 1998) and Hradec Kralove (since 1961, Vaniček et al., 2012). (line 57-58)

*L65: "designed" instead of "deigned"*
" The Dobson spectrophotometer is a double monochromator designed to measure $TCO_3$ " (line 67)

*Figure 2 and lines 90 – 95:*
*First of all, the authors should explain how equations (1) and (2) were derived. Were all data shown in Figure 2 used to derive these equations?*
" To eliminate a drift of the DS-ZS differences, the following transfer function from the regression line fit to the relative differences between DS and ZS $TCO_3$ subsets for $\mu \subset [2.8, 4.0]$) is used:" (line 97-99)

*Secondly, if the data shown in Figure 2 were used, then equation (2) has been calculated using a limited number of data points. Thus, I am not convinced that applying this relationship on future data would provide an accurate correction. Since data points for air mass above 4 are limited, and uncertainties in both the measurements of Dobson and MKII Brewer at such air masses are very large, I would recommend excluding data for air masses larger than 4 from the final, merged dataset.*
" However, the linear correction is not valid if $\mu > 4.0$ (Fig.2a). Only 0.7% of all $TCO_3$ observations had such high $\mu$ values. The fixed correction of 1.015 is applied if $\mu > 4.0$ but $TCO_{3, ZS}*$ values should be treated with caution." (line 104-106)

*Line 129: Please define R/N*
" The R/N table is used to convert the dial reading (the so-called R value) obtained by the Dobson observer into the logarithm of the ratio between the light intensities in a pair of the UV wavelengths with weak and strong absorption by ozone (the so-called N value). N values are used in theoretical formulas to calculate $TCO_3$ (e.g., Dobson, 1957). " (line 135-137)

*Section 2.2: Adding a Table summarizing the campaigns (place, reference instrument, etc) would be useful.*
Table 1 is added. (line 149)

*L172: Add reference(s) for the Brewer reference instrument. For example: Fioletov et al (2005).*

" Self-testing, PC-controlled instrument designed for continuous long-term observations in all weather conditions (e.g., Fioletov et al., 2005)." (line 178-179)

*L180: "This … spectrophotometers". Please rephrase. The meaning of this sentence is not clear.*

" The Dobson measurements provide instantaneous TCO$_3$ values, while the Brewer instrument gives the average of 5 observations, so this could be an additional source of differences between the spectrophotometers." (line 189-190)

*Figure 7: Even after the correction for the effective temperature there seems to be a trend in the ratio between the measurements from the two instruments (i.e. differences are ~+1% in 2002 – 2004 and ~-1% in 2018 - 2020). The authors should add some relative discussion (are these differences within the uncertainty of the merged dataset?).*

" The smoothed curved in Fig.7b provides that the Dobson TCO$_3$ values were ~1% lower (2002-2004) and ~1% higher (2018-2019) comparing with the Brewer values. Such discrepancies may be related to the Brewer ZS TCO$_3$ values, as they may be influenced by clouds (ZS Brewer algorithm is based on a statistical relationship with parallel DS observations), which in some years cause overestimation (or underestimation) in relation to the Dobson TCO$_3$ values." (line 223-227)

*Section 2.3.3: Discussion about the effect of stray light can be also found in: Moeini et al. (2019)*

" Moeini et al. (2019) discussed the differences between TCO$_3$ values measured almost simultaneously by the Dobson and Brewer spectrophotometers due to the stray light effect. They found that the difference for low solar elevations (slant TCO$_3$ > 800 DU) was related to the level of stray light withing the instruments, which is especially high for the single monochromator Brewer (Brewer Mark II), i.e., the same type as the Belsk's  Brewer " (line 241-244)

*In this latter paper the authors show that at very large ozone slant paths (i.e., for very large air masses) the role of stray light becomes exceptionally significant. That makes the measurements of both instruments unreliable. As I did earlier, I recommend again removing measurements for air masses larger than 4 from the analysis, as the uncertainties are already very large, solely due to the effect of stray light.*

" The present analysis show that the Brewer adjusted Dobson TCO$_3$ values are reliable for μ <4 or slant TCO$_3$ up to 1400 DU (Fig.10).  (line 353-354)

*Ideally, the authors should correct the measurements of both instruments for the effect of stray light, which of course is not a trivial task. Instead, they have scaled the measurements of Dobson to the measurements of the Brewer at large air masses. Assuming that the scaling is perfect, stray light still affects the measurements of the Brewer, and subsequently the ozone series. In any case, the authors should discuss, and try to quantify, the uncertainties related to the stray light effect.*

New Figure 10 is added illustrating the stray light effect.

" The correction for the stray light effects was applied to reduce the TCO$_3$ differences between the Dobson and Brewer instruments for low solar elevation. The correction is not calculated separately for each instrument. However, Figure 10 shows, that the Brewer Adjusted TCO$_3$ intraday values are only slightly sensitive to changes in slant TCO$_3$, i.e., within the max-min range between 0.99 and 1.01 derived from the smoothed profile of the ratio between TCO$_3$ values non affected (slant TCO$_3$ <800 DU) and affected by the stray light) (line 308-312)

---

## Author Response (AR1)

**Response to reviewer 1 comments and suggestions.**

*General comments:*

*The authors present one of the longest series of measurements of the total column of ozone globally. They also discuss the conditions and the procedures which ensure the high quality of the measurements. The scientific value of the presented dataset is high, and the manuscript is within the scope of the journal.*

*What I mainly miss, is a section wherein the authors would quantify the uncertainties of the final dataset. Uncertainty budget is of exceptional importance for anyone who would use the data. Thus, I strongly recommend that the authors should quantify the overall uncertainty and add the corresponding section.*

The new section is added " 2.4 Uncertainty of the Brewer adjusted Dobson $TCO_3$" describing uncertainties in the data. (line. 253-281 in the revised manuscript).

**2.4 Uncertainty of the Brewer adjusted Dobson $TCO_3$**

[revised manuscript text omitted]

*A few more changes are also necessary prior to the publication of the manuscript. Specific comments are provided below.*

**In response to the reviewer's comments, we present the text with the suggested correction. The numbering of the lines applied to the revised manuscript (not with marked changes).**

*In the data files, or at least in an accompanying description file, please specify whether time is UTC or something else*
We add that UTC time was used (see l.346)

The results of all intraday measurements for the period 1963-2019 were previously stored at the PANGAEA repository base with additional information including: time of observation (UTC), cloudiness type, air mass, and description of the wavelength pair and observation type selected for each individual measurement (Rajewska-Więch et al., 2020). The present analysis shows that the Brewer adjusted Dobson $TCO_3$ values are reliable for $\mu < 4$

*L7: please define that #84 is the serial number of the instrument.*
Yes. This is done in accordance with the reviewer's comment (see l.7)

**Abstract.** The total column ozone ($TCO_3$) measurements by the Dobson spectrophotometer #(serial No. 84) have been carried

*L13: please add "which were" before "also performed"*

Yes. This is done in accordance with the reviewer's comment (see l.13).

> values. To adjust this data to the Brewer spectrophotometer observations, which were also performed at Belsk, a procedure is

*L20: Please explain that #1 and #2 are the serial numbers of the instruments*

Yes. This is done in accordance with the reviewer's comment (see l.21-22).

> The monitoring of total column ozone (TCO₃) started in 1924 in Oxford (the United Kingdom) with prototype of the Dobson
> instrument (DI). Before the Second World War, there existed data records from 2 stations Oxford (DI #serial No. 1) and Arosa
> (DI #serial No. 2) archived in the World Ozone and Ultraviolet Radiation Data Centre (WOUDC).. After the international

*L22: "TCO" instead of "TOC" like in the rest of the document. At the same line, the authors probably mean that "the number of ozone observations increased sharply" instead of "The ozone observations were triggered".*

Yes. This is done in accordance with the reviewer's comment (see l.23-24)

> geophysical year in 1958, the total number of the stations with routine TOC₃TCO₃ increased up to about 50. The number of
> 25  ozone observations were triggeredincreased sharply in the early 1980s after following the discovery of the ozone hole in the

*L33: Delete "the"*

Yes. This is done in accordance with the reviewer's comment (see l. 34)

> duringimprovements in the 1970s, it was available for the ozone monitoring in the early 1980s. However, differences between

*L36: which ground-based network?*

We explain this in the revised manuscript: " in the TCO₃ ground-based observation network" (l.37-38)

> 1985), which replaced the previous ones by Vigroux (1967),) coefficients, have been used operationally in the TCO₃ ground-
> 40  based observation network since the early 1990s (Komhyr et al, 1993). Vanicek (2006) found that the difference could reach

*L42: Similar results to those reported by Redondas et al. (2014), have been also reported by Fragkos et al., (2015).*

Fragkos et al., (2015) paper has been added (see l.43).

> 45  coefficients significantly reduced the artificial seasonality in the Dobson-Brewer differences to less than 1% (Redondas et al.,
> 2014).; Fragkos et al., 2015). However, the data from the Dobson network has not yet been recalculated with use ofusing the

*L47:  In addition to Ball et al., the following study should be also cited:  Steinbrecht, et al. (2017).*

Steinbrecht et al. (2017) paper has been added (see l.47)

> stratospheric ozone in the NH midlatitudes,Steinbrecht et al., 2017; Ball et al., 2018). Moreover, In addition, a surprising

*L50: Please add references for the Arctic ozone depletion in 2020. For example: Wohltmann et al. (2020); Manney  et al. (2020); Inness et al. (2020).*

Papers by Wohltmann et al. (2020) and Manney et al. (2020) have been added (see l.53-54);

> antarctic-ozone-hole-large-and-deep?). Moreover, severe chemical losses occurred in the Arctic stratosphere in spring 2020
> (e.g., Manney et al., 2020; Wohltmann et al., 2020). Therefore, it is still worth monitoring ozone with the Dobson

*L54: Delete "of the"*

Yes. This is done in accordance with the reviewer's comment (see l. 56)

> Poland, started monitoring  atmospheric ozone (TCO₃ and the ozone vertical profile by the Umkehr method) on

*L55: Delete "including"*
Yes. This is done in accordance with the reviewer's comment (see l. 57-58)

>  March 23, 1963. There are only two stations in Europe with longer time series , Arosa (since 1926, Staehelin et al., 1998) and Hradec Kralove (since 1961, Vanicek et al., 2012). The importance of the Belsk's

*L65: "designed" instead of "deigned"*
Yes. This is done in accordance with the reviewer's comment (see l. 67)

> The Dobson spectrophotometer was designed to measure TCO₃ by the technique of the

*Figure 2 and lines 90 – 95:*
*First of all, the authors should explain how equations (1) and (2) were derived. Were all data shown in Figure 2 used to derive these equations?*
In the revised manuscript, we explain that the regression fit was used to part of the data with μ in the range [2.8,4.0] (see l. 96-97).

> types and their positions in the sky. Therefore, the DS-ZS differences can sometimes exceed 5%. To eliminate the drift of the DS-ZS differences, the following transfer function is used from the regression line fit to the relative differences between DS and ZS TCO₃ subsets for $\mu \subset [2.8, 4.0]$):

*Secondly, if the data shown in Figure 2 were used, then equation (2) has been calculated using a limited number of data points. Thus, I am not convinced that applying this relationship on future data would provide an accurate correction. Since data points for air mass above 4 are limited, and uncertainties in both the measurements of Dobson and MKII Brewer at such air masses are very large, I would recommend excluding data for air masses larger than 4 from the final, merged dataset.*
In the revised text we explain that the number of observations with μ > 4 was small and we decided to keep such data but should be treated with caution (see l.102-104)

> is pertaining the corrected value for μ >2.8. However, the linear correction is not valid if μ > 4.0 (Fig.2a). Only 0.7% of all TCO₃ observations were made at such high μ values. The fixed correction of 1.015 is applied if μ > 4.0 but $TCO_{3, ZS}$* values
> 115 should be treated with caution. The smoothed pattern of the DS-ZS differences (Fig.2b) is close to zero after the application

*Line 129: Please define R/N*

In the revised manuscript, we explain the meaning of R/N (l. 130-135).

> calculation of the resulting R/N tables were of key importance for maintaining the quality of the Belsk's Dobson. The R/N table is used to convert the dial reading (the so-called R value) obtained by the Dobson observer into the logarithm

> of the ratio between the light intensities in a pair of the UV wavelengths with weak and strong absorption by ozone (the so-called N value). N values are used in theoretical formulas to calculate TCO₃ (e.g., Dobson, 1957).

*Section 2.2: Adding a Table summarizing the campaigns (place, reference instrument, etc) would be useful.*
Table 1 has been added. (see l. 147)

**Table 1. The intercomparison campaigns with the Dobson instrument from Belsk.**

| Site/Country | Year | Standard Instrument |
|---|---|---|
| Belsk/Poland | 1974 | World Standard. Dobson No.83 |
| Potsdam/Germany | 1979 | World Standard. Dobson No.83 |
| Arosa/Switzerland | 1986 | World Standard. Dobson No.83 |
| Arosa/Switzerland | 1990 | World Standard. Dobson No.83 |
| Arosa/Switzerland | 1995 | World Standard. Dobson No.83 |
| Hohenpeissenberg/Germany | 2001 | European Sub-Standard. Dobson No.64 |
| Hohenpeissenberg/Germany | 2005 | European Sub-Standard. Dobson No.64 |
| Hohenpeissenberg/Germany | 2009 | European Sub-Standard. Dobson No.64 |
| Hohenpeissenberg/Germany | 2014 | European Sub-Standard. Dobson No.64 |

*L172: Add reference(s) for the Brewer reference instrument. For example: Fioletov et al (2005).*
Fioletov et al (2005) paper has been added (see l. 175)

195   (BS64) Mark II ( single monochromator) was launched at Belsk in 1991. Like other Brewers, the BS64 is a fully automated, self-testing, PC-controlled instrument intended for continuous, long-term observations in all weather conditions (e.g., Fioletov et al., 2005). The quality of BS64 measurements has been supported by regular (yearly or

*L180: "This … spectrophotometers". Please rephrase. The meaning of this sentence is not clear.*
In the revised text, we explain that  the difference concerns the  Dobson and Brewer data (see l. 184-186).

of these $TCO_3$ values is small. i.e., standard deviation is less than 2.5 Dobson unit (DU). The Dobson spectrophotometer provides instantaneous $TCO_3$ values, while the Brewer instrument gives the average of 5 observations, so this may be an additional source of differences between the spectrophotometers.

*Figure 7: Even after the correction for the effective temperature there seems to be a trend in the ratio between the measurements from the two instruments (i.e., differences are ~+1% in 2002 – 2004 and ~-1% in 2018 - 2020). The authors should add some relative discussion (are these differences within the uncertainty of the merged dataset?).*
However, the trend calculated in the Brewer-Dobson differences after the correction for the ozone absorption coefficients dependence to temperature is not statistically significant for the period 2002-2019. In the added text, we explain possible sources of the differences in the periods 2002-2004 and 2018-2020. (see l.217-220)

3% (ZS). The smoothed curved in Fig.7b provides that the Dobson $TCO_3$ values were ~1% lower (2002-2004) and ~1% higher (2018-2019) comparing with the Brewer values. Such discrepancies may be related to the Brewer ZS $TCO_3$ values, because they may be influenced by clouds (the Brewer ZS algorithm is based on a statistical relationship with parallel DS observations), which in some years result in overestimation (or underestimation) in relation to the Dobson $TCO_3$ values.

*Section 2.3.3: Discussion about the effect of stray light can be also found in: Moeini et al. (2019)*
In the revised manuscript, we discuss results by this paper. (see l.234-238)

2015). Moeini et al. (2019) discussed the differences between TCO$_3$ values measured almost simultaneously by the Dobson and Brewer spectrophotometers due to the stray light effect. They found that the difference for low solar elevations (slant TCO$_3$ > 800 DU) was related to the instrument's individual sensitivity to stray light, which may be particularly high for a single monochromator Brewer (Brewer Mark II), i.e., the same type operating at Belsk.

*In this latter paper the authors show that at very large ozone slant paths (i.e., for very large air masses) the role of stray light becomes exceptionally significant. That makes the measurements of both instruments unreliable. As I did earlier, I recommend again removing measurements for air masses larger than 4 from the analysis, as the uncertainties are already very large, solely due to the effect of stray light.*

We are aware of the stray light problem in the Brewer and Dobson spectrophotometers, so we propose the correction function for this effect. (l.300-303)

The last step of the homogenization was to eliminate the differences between the spectrophotometers found in periods of low solar elevation. These are probably related to the presence of stray light in spectrophotometers, causing underestimation of TCO$_3$ values at low solar elevations. The correction for the differences in the stray light effect in the spectrophotometers was

340     proposed, see function (3), to reduce the B-D TCO$_3$ differences for low solar elevation. The stray light correction was not

In the conclusion section, we mention the following limitations on the use the merged time series (see l. 347-348)

2020). The  present analysis shows that the Brewer adjusted Dobson TCO$_3$ values are reliable for μ <4 or slant TCO$_3$ up to 1400 DU (Fig.10).

*Ideally, the authors should correct the measurements of both instruments for the effect of stray light, which of course is not a trivial task. Instead, they have scaled the measurements of Dobson to the measurements of the Brewer at large air masses. Assuming that the scaling is perfect, stray light still affects the measurements of the Brewer, and subsequently the ozone series. In any case, the authors should discuss, and try to quantify, the uncertainties related to the stray light effect.*

True, the Brewer total column ozone was not corrected for the stray light effects. We propose the correction of the Dobson ozone to fit as close as possible the present Brewer data (without correction for the stray light). Figure 10 shows the Brewer adjusted Dobson column ozone is only slightly sensitive to the stray light. (l. 303-306)

340     proposed, see function (3), to reduce the B-D TCO$_3$ differences for low solar elevation. The stray light correction was not calculated separately for each instrument. However, Figure 10 shows, that the Brewer Adjusted TCO$_3$ values are only slightly sensitive to changes in slant TCO$_3$, i.e., within the max-min range between 0.99 and 1.01 derived from the smoothed profile of the ratio between TCO$_3$ values non affected (slant TCO$_3$ <800 DU) and affected by the stray light.

**Response to reviewer 2 comments and suggestions.**

*This paper aims to provide a documentary of the long-term total ozone measurements at Belsk, Poland. This paper is well written and provides a great deal of details about record homogenization and calibration. I have studied stratospheric and tropospheric ozone variabilities for a long time, this manuscript fills me with some measurement history. In terms of data documentation, the material and presentation of the paper is nearly impeccable.*

*Whereas the authors point out the unexpected CFC emission and 2020 Antarctic ozone hole in the introduction, these issues are not discussed anywhere after the introduction. Since in the research community, the current mainstream seeks to address trends and variability attribution at detailed vertical structure/pressure surfaces, the total ozone measurements are rather handcuffed to answer the questions from a broader perspective. But the CFC emission and 2020 Antarctic ozone hole should be at least discussed further, for example, Belsk is a high latitude location, are the measurements affected by the Antarctic ozone hole in spring of 2020? As far as I recall, I have seen that the impact can be observed by Canadian ozonesonde records.*

In the introduction, we discussed the unexpected upward trend in CFC emissions in recent years, emphasizing the need to continue observing total ozone in the world.

This topic is not discussed later in the text, as manuscripts submitted to the journal should mainly focus on data description and procedures supporting the data quality. Therefore, the reviewer's suggestion to discuss the effect of the Antarctic ozone hole on total ozone in the mid-latitude NH using data from Belsk is left for further consideration.